# Can parasites halt the invader? Mermithid nematodes parasitizing the yellow-legged Asian hornet in France

Claire Villemant[1], Dario Zuccon[2], Quentin Rome[1], Franck Muller[1], George O. Poinar Jr[3] and Jean-Lou Justine[1]

[1] Institut de Systématique, Évolution, Biodiversité, ISYEB, UMR 7205 – CNRS, MNHN, UPMC, EPHE, Muséum National d'Histoire Naturelle, Sorbonne Universités, Paris, France
[2] Service de Systématique moléculaire, UMS 2700 CNRS, Muséum National d'Histoire Naturelle, Paris, France
[3] Department of Integrative Biology, Oregon State University, Corvallis, OR, USA

Corresponding author
Claire Villemant,
villeman@mnhn.fr

## ABSTRACT

Since its introduction in France 10 years ago, the yellow-legged Asian bee-hawking hornet *Vespa velutina* has rapidly spread to neighboring countries (Spain, Portugal, Belgium, Italy, and Germany), becoming a new threat to beekeeping activities. While introduced species often leave behind natural enemies from their original home, which benefits them in their new environment, they can also suffer local recruitment of natural enemies. Three mermithid parasitic subadults were obtained from *V. velutina* adults in 2012, from two French localities. However, these were the only parasitic nematodes reported up to now in Europe, in spite of the huge numbers of nests destroyed each year and the recent examination of 33,000 adult hornets. This suggests that the infection of *V. velutina* by these nematodes is exceptional. Morphological criteria assigned the specimens to the genus *Pheromermis* and molecular data (18S sequences) to the Mermithidae, due to the lack of *Pheromermis* spp. sequences in GenBank. The species is probably *Pheromermis vesparum*, a parasite of social wasps in Europe. This nematode is the second native enemy of *Vespa velutina* recorded in France, after a conopid fly whose larvae develop as internal parasitoids of adult wasps and bumblebees. In this paper, we provide arguments for the local origin of the nematode parasite and its limited impact on hornet colony survival. We also clarify why these parasites (mermithids and conopids) most likely could not hamper the hornet invasion nor be used in biological control programs against this invasive species.

Subjects Agricultural Science, Entomology, Environmental Sciences, Parasitology, Zoology
Keywords Invasive species, Asian hornet, France, Nematodes, Biological control, Hymenoptera

## INTRODUCTION

The recent introduction of the Yellow-legged Asian hornet *Vespa velutina* in France was the first successful invasion of an exotic Vespidae in Europe (*Rasplus et al., 2010*; *Beggs et al., 2011*). This species is of great concern among public authorities and beekeepers because of its rapid multiplication and high impact on beekeeping due to its predatory action on honeybees (*Perrard et al., 2009*) and its hawking behavior that disrupts bee colony foraging (*Rortais et al., 2010*; *Monceau et al., 2013*; *Arca et al., 2014*). This invasive hornet was first

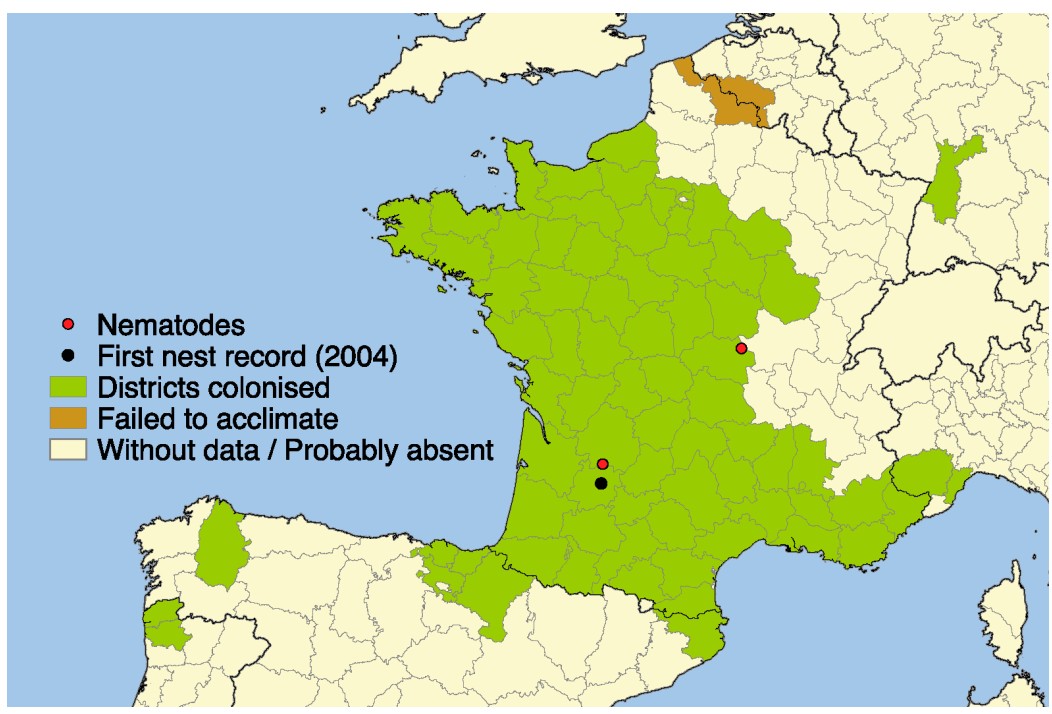

**Figure 1 Current distribution of *Vespa velutina*.** Distribution of the invasive yellow-legged Asian hornet *Vespa velutina* in Europe in 2014. Black spot: first occurrence of *V. velutina* in Europe. Red spots: localities where the nematodes have been found.

observed in 2004 in Southwest France (*Villemant et al., 2011*); since then it has spread out across 67 French departments (ca. 360,000 km$^2$) (*Rome et al., 2013*; *INPN, 2015*). In addition, it spread to Spain in 2010, to Portugal and Belgium in 2011 (*Rome et al., 2013*), to Italy in 2012 (*Demichelis et al., 2014*), and arrived in Germany in 2014 (R Witt, pers. comm., 2014) (Fig. 1). It is expected to eventually spread throughout Europe (*Villemant et al., 2011*) and with recent climate change scenarios, future range expansion may be even more rapid than in the past ten years (*Barbet-Massin et al., 2013*).

Multiple biotic factors, including resources, competition, and natural enemies can affect the demographics of an invader, either independently or interactively and thus play a role in its establishment. Introduced species often leave behind natural enemies from their original home, thus benefiting them in the new environment and resulting in increased growth, reproduction and competitive ability (*Holway, Suarez & Case, 1998*; *Colautti et al., 2004*; *Torchin & Mitchell, 2004*; *Lee & Klasing, 2004*; *Roy et al., 2011*). Local recruitment of natural enemies in their new home can also affect the development of the invasion and lessen their effect on local hosts (*Prenter et al., 2004*; *Girardoz, Kenis & Quicke, 2006*; *Dunn, 2009*; *Kenis et al., 2009*; *Péré et al., 2011*).

The first native enemy of *Vespa velutina* reported in Europe was a thick-headed fly (family Conopidae) whose larvae develop as internal parasitoids of adult wasps and bumblebees (*Darrouzet, Gévar & Dupont, 2014*). We report here a new parasite, a mermithid nematode of the genus *Pheromermis* that was obtained from *V. velutina* adults

in 2012 in two different French localities. As far as we know, no other nematode parasites of *V. velutina* have been reported up to now in Europe.

In this paper, we discuss how these parasites could potentially hamper the hornet invasion and whether they could be used in biological control programs against this invasive species.

## MATERIALS AND METHODS

### Origin of specimens

The invasive progress of the alien hornet has been monitored since 2006 through an online biodiversity database maintained by the Muséum National d'Histoire Naturelle (MNHN) and regularly updated by one of us, Q.R. (*Rome et al., 2013*; *INPN, 2015*). This monitoring showed that more than 7,000 nests were discovered from 2006 to 2014. Nests are mainly observed in autumn after leaf fall, when the colonies reach maturity and contain several hundred to two thousand adult hornets (*Rome et al., 2015*). This surveillance network provides useful information but the hornets were not regularly surveyed for parasites. However, in order to study seasonal changes in *V. velutina* colony structure, we dissected 77 nests from four early invaded French districts (mainly Dordogne and Gironde) between 2007 and 2011 (*Rome et al., 2015*). Nests kept frozen at −25 °C were thawed for dissection and some 33,000 adult hornets sorted, manipulated and weighed. Such manual handling did not lead to discovery of any individual having a burst, distended or flaccid abdomen demonstrating the presence of a nematode. Hornets were not dissected but if some of them had been infected, the presence of nematodes would have been detected. Indeed, at the end of its development, the parasitic nematode entirely fills the host's abdomen, whose segments can easily separate from each other after freezing and thawing. This was the case for the only parasitized adult we obtained which contained a conopid pupa that has not been identified (*Villemant et al., 2008*).

Nematode parasites were unexpectedly noticed in hornets by local observers on two occasions. In November 2012, one mermithid was obtained from ten adult hornets dissected from a nest collected at Dompierre-sur-Besbre, Allier (P Noireterre, pers. comm., 2012). In January 2013, two mermithids were obtained from dead adults in an advanced state of decomposition, from a nest at Issigeac, Dordogne (JP Doumenjou-Larroque, pers. comm., 2013). The mermithids were sent to the MNHN for identification.

### Morphology

The mermithid nematode from Dompierre-sur-Besbre (Allier) was photographed (Fig. 2) and basic measurements (length, width) were taken; then part of its body was sampled for the molecular study, and the remainder was submitted to one of the authors (G.O.P.) for further identification. Mermithid nematodes, including the present species, are quite large at maturity and often exceed the length of their host. The specimen from the Asian hornet was a postparasitic juvenile and had morphological characters that aligned it with species of the genus *Pheromermis* (*Poinar, Lane & Thomas, 1976*). The specimens are deposited in the MNHN collection as MNHN JL50 (Dompierre-sur-Besbre) and MNHN JL51A and JL51B (Issigeac).

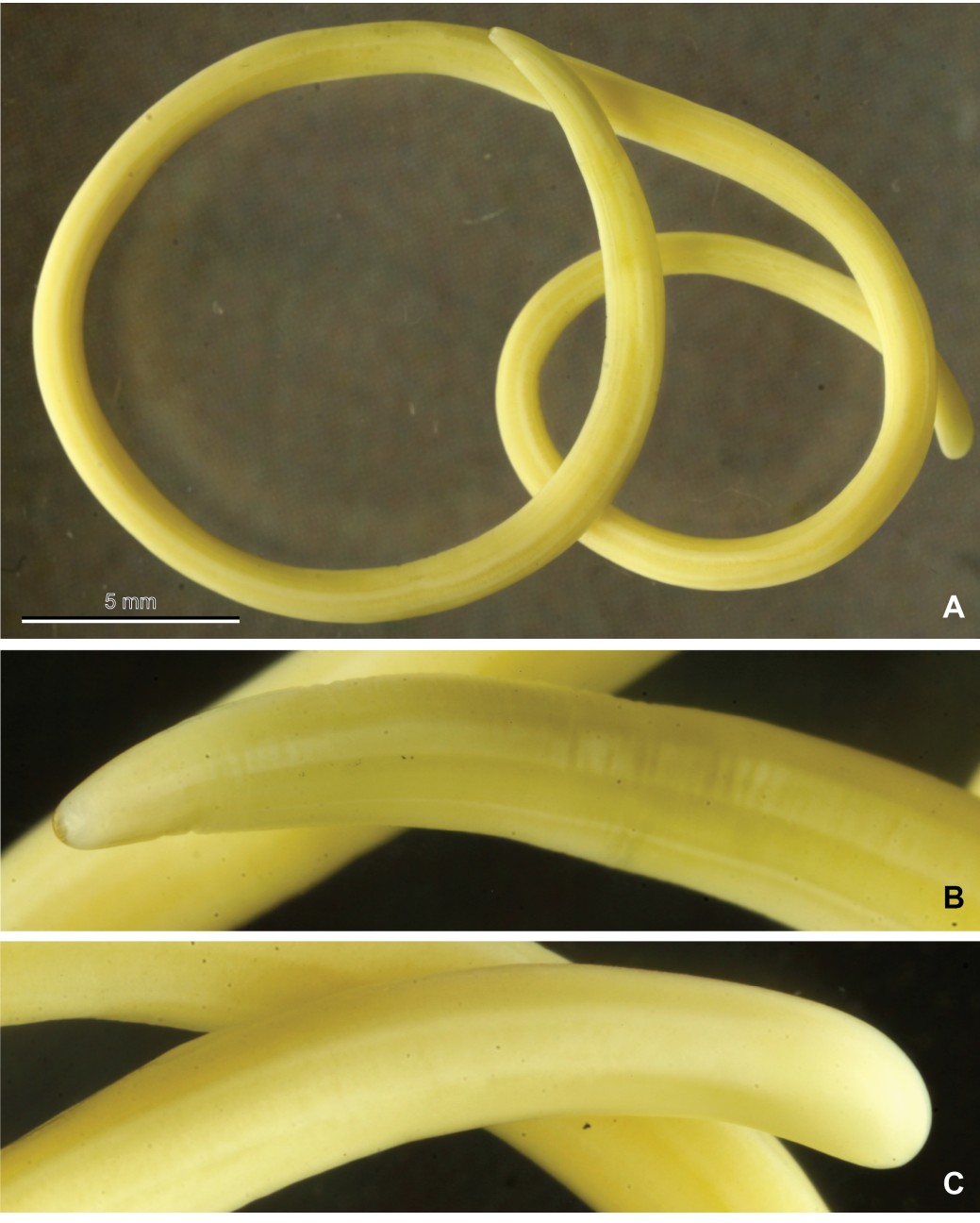

**Figure 2 Mermithid Nematode from *Vespa velutina*.** Photographs of a postparasitic juvenile of the mermithid nematode (MNHN JL50) from *Vespa velutina*, collected in Dompierre-sur-Besbre, France. (A), whole worm; (B), head; (C), tail.

## Molecular identification

Total genomic DNA was extracted from a 5 mm long medial segment sampled from each specimen, using the Qiagen DNA Mini Kit and following the manufacturer's protocol. Three candidate genes were selected for PCR amplification: the mitochondrial cytochrome oxidase I (COI) and the nuclear large and small subunit rRNA genes (28S-rRNA and

Table 1 **Primers used.** Primer pairs used in this study with their annealing temperatures. The 18S was amplified in two overlapping fragments.

| Gene | Primers | | Annealing T | Reference |
|------|---------|--|-------------|-----------|
| 18S | 18S-1F | TACCTGGTTGATCCTGCCAGTAG | 51 | *Giribet et al. (1996)* |
| | 18S-5R | CTTGCAAAGCTGCTTTCGC | | |
| | 18S-3F | GTTCGATTCCGGAGAGGGA | 51 | |
| | 18S-Bi | GAGTCTCGTTCGTTATCGGA | | |
| 28S | 28S-C1 | ACCCGCTGAATTTAAGCAT | 55 | *Dayrat et al. (2001)* |
| | 28S-D2 | TCCGTGTTTCAAGACGGG | | |
| COI | AnCO1-F | ATTTGGTCTTTGATCTGGTATGG | 48 | *Cross et al. (2006)* |
| | AnCO1-R | TGGCAGAAATAACATCCAAACTAG | | |
| | LCO1490 | GGTCAACAAATCATAAAGATATTGG | 48 | *Folmer et al. (1994)* |
| | HCO2198 | TAAACTTCAGGGTGACCAAAAAATCA | | |

18S-rRNA). The choice was dictated by the large number of nematode sequences already available in GenBank for comparison.

The genes were amplified using standard primers and amplification profiles (Table 1). The PCRs were conducted in 20 µl reaction volume, containing 1–5 ng of DNA and to a final concentration of 1× reaction buffer, 2.5 mM MgCl2, 0.26 mM dNTP, 0.3 µM of each primer, 5% DMSO and 1.5 units of Qiagen Taq polymerase. For all primer pair combinations, the amplification profile was: 5 min initial denaturation at 94 °C, 40 cycles of 40 s at 94 °C, 40 s at primer annealing temperature (see Table 1) and 60 s at 72 °C, followed by a final extension of 5 min at 72 °C. PCR products were visualized on a 1.5% agarose gel stained with ethidium bromide and the positive PCRs were sequenced in both directions using the Sanger method.

A preliminary BLAST search suggested that the 18S and 28S sequences obtained from the samples were rather similar to those of Mermithidae present in GenBank. However, only two out of the twelve 28S GenBank sequences overlapped significantly with our sequences, so further comparisons were restricted to the 18S gene. All 18S sequences of Mermithidae available in GenBank were then downloaded and aligned with the sequences we obtained, resulting in a 1,322 bp alignment. The 18S GenBank sequences proved to be rather variable in length, some of them with minimal overlap with the region amplified using our primers. The dataset was then reduced, retaining only the sequences longer than 400 bp.

New sequences have been submitted to GenBank under the accession numbers KR029620 and KR029622 (MNHN JL50) and KR029621 and KR029623 (MNHN JL51A).

The final matrix, including 26 ingroup sequences and five outgroups (Aulolaimidae: *Aulolaimus* and Isolaimiidae: *Isolaimium*) was analyzed under the maximum likelihood criteria, using RAxML v. 7.0.3 (*Stamatakis, 2006*), selecting a GTR+$\Gamma$+I model and random starting tree, with empirical base frequencies and estimated $\alpha$-shape parameters and GTR-rates. Nodal support was estimated using 100 bootstrap replicates.

## RESULTS

### Morphology

Only the specimen from Dompierre-sur-Besbre revealed morphological features (Fig. 2); it was 81 mm in length with a maximum width of 1.3 mm. The other two specimens from Issigeac were too damaged by putrefaction. The morphological features of the single juvenile examined were consistent with those of juveniles of the European wasp mermithid, *Pheromermis vesparum* (*Kaiser, 1987*), a species specialized on social vespids.

The genus *Pheromermis* is characterized by the presence of four submedian cephalic papillae; large anteriorly placed cup-shaped amphids; an S-shaped vagina not bent in a transverse plane to the body; six hypodermal cords; paired, short, separate spicules; cuticle with cross fibers; and eggs lacking processes (*Poinar, Lane & Thomas, 1976*). Because most of these are characters of the adults and the specimens from the Asian hornet were postparasitic juveniles, it is possible that more than one species of *Pheromermis* are involved.

### Molecular identification

Of the three specimens analyzed, one from Issigeac was probably too decomposed and the resulting DNA likely too degraded because all amplification attempts failed. Instead we recovered sequences for the 18S and 28S genes from the other two specimens i.e., from two localities. The 18S sequences were identical, while the 28S differed only by 1.6%, with all variable positions concentrated in the loop regions. Despite several attempts, the amplification of the COI gene also failed, suggesting that the standard primers are ineffective for amplifying the COI gene in this species.

The tree obtained from the maximum likelihood analysis recovers the two 18S sequences well nested within the family Mermithidae, and sisters to three specimens identified as *Mermis nigrescens* (Fig. 3), a parasite of grasshoppers (*Baker & Capinera, 1997*). These results are congruent with the morphological identification, although no molecular data of *Pheromermis* are currently available for comparison and thus, we cannot identify the specimens at the species level. We have to consider them as belonging to *Pheromermis* sp. However, larvae collected from *V. velutina* most probably belong to *Pheromermis vesparum* (*Kaiser, 1987*), a well-known parasite of social wasps. It has been recorded from *Vespa crabro*, *Vespula vulgaris*, *V. germanica*, *Dolichovespula saxonica* and *Polistes* sp. (*Kaiser, 1987*).

## DISCUSSION

In a ten year span, only 3 nematodes have been collected from hornets in two distant localities in France. Nevertheless, numerous and very populous colonies of *V. velutina* are destroyed and dropped from tree crowns each year in France. Dead hornets spill onto the ground and many of them are crushed during the operation. In spite of these huge numbers of destroyed nests and manipulated hornets, very few worms that have such a large size were detected. This suggests that the infection of the hornet by these nematodes is exceptional.

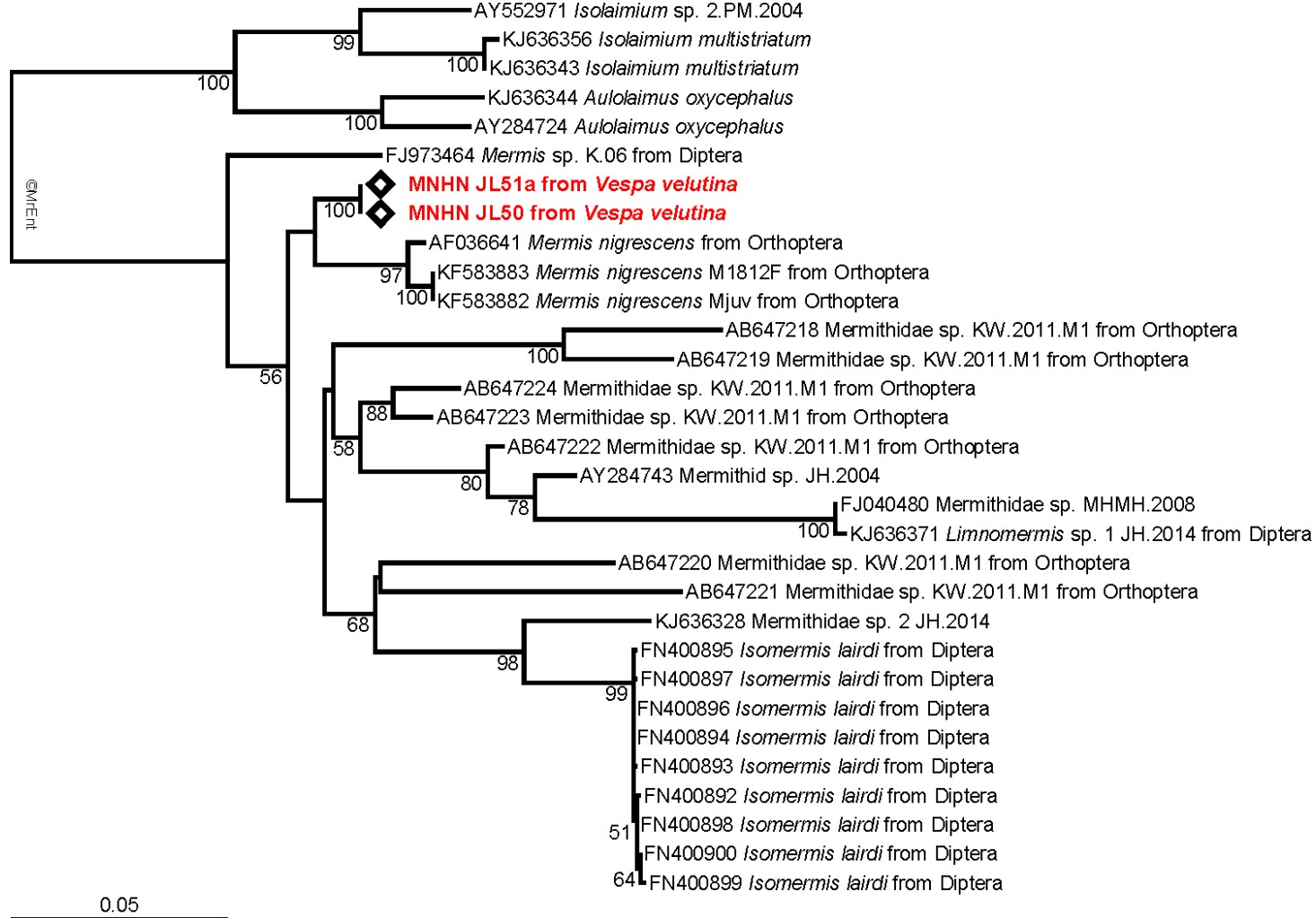

**Figure 3 Molecular analysis.** Maximum-likelihood 18S tree showing the relationships of the parasite infecting *Vespa velutina* (red) with other mermithids. When available, the main host insects are indicated after the parasite name. The bootstrap support values are indicated at the node.

The development of the *Pheromermis* species is unique among the Mermithidae because a second host (paratenic or transport host) is required for life cycle completion (*Poinar, Lane & Thomas, 1976*; *Kaiser, 1987*; *Martin, 2004*). The adult nematodes occur in water or saturated soils and the eggs are fully embryonated at oviposition. The eggs hatch in the gut of various aquatic or semi-aquatic insects and infective juvenile stages penetrate the gut wall to enter a quiescent state in the tissues of the paratenic hosts, even during host metamorphosis to adult form. Wasp larvae are probably infected when they are fed with adult paratenic hosts captured by worker wasps. The nematode larvae become active and start feeding on non-vital tissues of the developing wasp. This coincides with the period when social wasps are raising their sexual brood (*Kaiser, 1987*). As nematodes rarely kill their juvenile wasp hosts, the adult wasp emerges and the nematode matures in the abdomen of the wasp, rendering sexual individuals sterile or inactive. When the infected wasp visits water in the fall (before the future queens enter hibernation), the mature worm

leaves its host, which kills it, molts into the adult stage, mates and lay eggs, so completing the life cycle. It is not yet known whether the reproductive wasps normally visit free water areas or whether the mermithids cause hosts to seek water (*Poinar, 1976*).

Two hypotheses can be made to explain the presence of the nematode parasites in *V. velutina* adults. The parasite (1) may have been introduced in the new range by the invader itself, or (2) may have been acquired from the local fauna. The first hypothesis is unlikely because hornet queens parasitized at the time of their introduction would have died without descendants. Moreover, genetic data have shown that only very few hornet queens (or even a single individual) have been introduced in France (*Arca et al., 2015*), therefore making the hypothesis of an introduced parasite even less probable. In addition, the exotic parasite would have had to adapt to one (or several) local insect species whose larvae are aquatic. The second hypothesis seems to be more likely. The invasive species was infested by an autochthonous nematode whose paratenic hosts are various aquatic insects and main hosts are autochthonous social wasps. *Pheromermis* species that attack social wasps in Europe have a wide range of hosts (*Molloy, Vinikour & Anderson, 1999*) and thus are more likely to infect a new host than more specific nematodes.

Known paratenic hosts of *Pheromermis* spp. attacking wasps include notably larvae of caddisflies (Trichoptera), stoneflies (Plecoptera), craneflies (Tipulidae) and mayflies (Ephemeroptera), as well as various Coleoptera larvae (*Poinar, Lane & Thomas, 1976*; *Poinar, 1981*; *Molloy, Vinikour & Anderson, 1999*). In the course of an ongoing study, we collected more than 2000 prey flesh pellets at the time they were brought back to the nest by worker wasps. Their identification showed that among potential paratenic hosts, only caddisflies are part of the *V. velutina* prey spectrum and only in tiny proportions (0.2%) compared to other insect preys (unpublished data from the authors).

Local recruitment of natural enemies like mermithid nematodes obviously leads to the following question: are they able to control hornet populations? The fact that *Pheromermis* spp. kill their hosts makes these nematodes important as biological control agents of social wasps. However, *Martin (2004)* noted that contrary to the 50% quoted by *Poinar, Lane & Thomas (1976)* and *Moller et al. (1991)*, levels of infection by *Pheromermis* spp. are actually lower and vary from 0–7% in workers and males, and 8–35% in social wasp future queens (*Blackith & Stevenson, 1958*; *Kaiser, 1987*). In 1893, all males in one large *Vespula* wasp nest were found infected (*Fox-Wilson, 1946*) by *Gordius* worms (*Gordius* belong to the Nematomorpha, a phylum distinct from the Nematodes to which mermithid belong, but with a similar life cycle). However, the dissections of thousands of adults from hundreds of *Vespula* and *Vespa* nests by various wasp researchers indicate that such extreme levels of infection are very rare (*Martin, 2004*). Our extensive survey (*Rome et al., 2015*) suggests the same conclusion.

The degree of infection in any nest also depends on the proximity of the wasp nest to an abundant source of the nematodes' paratenic hosts (*Rose, Harris & Glare, 1999*; *Martin, 2004*). *Kaiser (1987)* found that 1/3 of *Vespula* nests were infected with *Pheromermis* when, and only when, they were within 200 m of water. Also, the rarity of potential paratenic hosts in *V. velutina*'s prey spectrum would not enable the nematode to greatly infest a

colony at high densities, even if hornet workers generally fed several larvae with every flesh pellet brought back to the nest (*Janet, 1903*; *Spradbery, 1973*). Moreover, a high mortality of nestmates is not sufficient to ensure the total destruction of a colony, and even with 75% mortality, recovery is possible (*Gambino, Pierluisi & Poinar, 1992*; *Toft & Harris, 2004*; *Gouge, 2005*). The mother queen that founded the colony also cannot be infected since the nematode infests its host at the larval stage and kills it in the fall; infested female sexual adults die before funding a colony. The maturing nematode, which eventually occupies all the gaster when the sexual stages emerge, severely interferes with the amount of fat deposited (*Martin, 2004*). Finally, unlike many entomopathogenic nematodes, *Pheromermis* spp. do not seem to serve as vectors for symbiotic insect-pathogenic bacteria (*Poinar, 1979*), whose presence may increase the virulence of the infection (*Lacey et al., 2001*).

The possible use of *Pheromermis* spp. as biological control agents against social wasps was tested with a simulation model (*Martin, 2004*) which predicted that the production of sexual individuals would be reduced in colonies undergoing early and high levels of infection. However, even highly infected (80%) colonies can still produce some reproductive offspring, indicating that they are resilient to infection. Moreover, an increase of the infestation level raises the larva/worker ratio so that less fed larvae produce sexual females of lesser quality, with fewer over-wintering and nest founding successes (*Harris & Beggs, 1995*). The low quality of some sexual females may inadvertently permit a greater founding success via density-dependent compensation: the healthy females are less numerous but they experience lower mortality due to weaker competition for fat storage (*Harris & Beggs, 1995*) and reduced nest usurpation in the spring (*Martin, 1991*; *Martin, 2004*; *Archer, 2012*).

The reduction of usurpation disputes which usually lead to the death of a high number of founder queens (*Spradbery, 1991*; *Archer, 2012*) also explains why conopid flies would not be efficient control agents even if they are able to directly attack funder queens. Thus, the local recruitment of *Conops vesicularis* as a parasitoid of *Vespa velutina* in France (*Darrouzet, Gévar & Dupont, 2014*) would not make it a potential control agent of the invasive hornet. Moreover, conopids mainly fly in summer from June to September (*Schmid-Hempel et al., 1990*) and are thus more likely to attack foraging workers than mother queens, which do not leave the nest after their first workers emerge in June (*Matsuura & Yamane, 1990*; *Rome et al., 2015*).

Like *Pheromermis* spp., many parasitoids of social wasps, such as the ichneumonids *Sphecophaga* spp. (*Donovan et al., 2002*; *Beggs et al., 2008*), the stylops *Xenos* spp. (*Matsuura & Yamane, 1990*), or the conopid flies like *C. vesicularis*, attack only single individuals within the colony. By contrast, other parasites such as *Varroa destructor* mites can kill a bee colony of more than 30,000 individuals by transmitting viral pathogens as they move between bees within a colony (*Sumpter & Martin, 2004*). Infection levels of parasites which attack only single individuals need to be very high ($>50\%$) to kill or significantly reduce the productivity of social wasp colonies (*Matsuura & Yamane, 1990*; *Barlow, Beggs & Barron, 2002*), because their populations have high reproductive efficiency and undergo density-dependent compensation in spring (*Martin, 1991*; *Martin, 2004*).

On another issue, the introduction of an alternative (invading) host, rather than diluting the effects of a parasite, may act as a reservoir for infection, a factor exacerbated by high densities of the invading hosts. The arrival of an alternative host could thus favor the multiplication of the native parasitoid, resulting in reduced population growth of susceptible hosts (*Holt & Lawton, 1993*). Such indirect host competition can lead to the extinction of the most parasitized host (*Prenter et al., 2004*; *Dunn, 2009*). Nematodes are considered to have limited effects on social wasps (*Gouge, 2005*) whereas conopid flies which rarely attack social wasps (*Spradbery, 1973*; *Matsuura & Yamane, 1990*) can locally be extremely destructive to bumblebee colonies in Europe (*Schmid-Hempel, 2001*). The negative effects of parasitoids on their hosts are however not always clear-cut and immediately visible. Parasitoid effects often depend on host condition and may only be expressed when the host population is in poor condition (*Schmid-Hempel, 2001*), a threat which nowadays may apply more to bumblebees whose populations are more susceptible to decline (*Gillespie, 2010*) than those of social wasps.

## ACKNOWLEDGEMENTS

We are most thankful to Philippe Noireterre and Jean-P Doumenjou-Larroque, the two persons who collected the nematodes, Claire Ménissier from the journal "La Hulotte" who put us in touch with the second collector, as well as to all persons that provided nests for this study. We also thank the two anonymous reviewers for their helpful comments on the manuscript.

### Funding

This work was supported by the MEDDE (Ministère de l'Écologie, du Développement Durable et de l'Énergie), and the NAAS-RDA KOREA (National Academy of Agricultural Science of the Rural Development Administration of the Republic of Korea; project Ecology and integrated control of *Vespa velutina*, 2013–2014). The salary of Dario Zuccon was covered by a grant from the thematic action of the Muséum National d'Histoire Naturelle "Taxonomie moléculaire: DNA Barcode et gestion durable des collections." The funders had no role in study design, data collection and analysis, decision to publish, or preparation of the manuscript.

### Grant Disclosures

The following grant information was disclosed by the authors:
MEDDE.
NAAS-RDA KOREA.

### Competing Interests

Jean-Lou Justine is an Academic Editor for PeerJ.

## Author Contributions

- Claire Villemant and Franck Muller conceived and designed the experiments, performed the experiments, analyzed the data, wrote the paper, reviewed drafts of the paper.
- Dario Zuccon analyzed the data, contributed reagents/materials/analysis tools, wrote the paper, prepared figures and/or tables, reviewed drafts of the paper, did molecular analyses.
- Quentin Rome and Jean-Lou Justine conceived and designed the experiments, performed the experiments, analyzed the data, prepared figures and/or tables, reviewed drafts of the paper.
- George O. Poinar Jr reviewed drafts of the paper, examined the morphology of nematode specimens.

## DNA Deposition

The following information was supplied regarding the deposition of DNA sequences: GenBank numbers KR029620–KR029623.

## Supplemental Information

Supplemental information for this article can be found online at http://dx.doi.org/10.7717/peerj.947#supplemental-information.

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
