# Peer review of "Can parasites halt the invader? Mermithid nematodes parasitizing the yellow-legged Asian hornet in France"

_PeerJ, doi:10.7717/peerj.947_

## Round 0.1 · original submission · Minor Revisions

Thank you for your submission to PeerJ. I am writing to inform you that your manuscript requires some very minor revisions before acceptance.

Reviewer 1 ·

Basic reporting

No comments

Experimental design

No comments

Validity of the findings

No comments

Additional comments

This manuscript reports a newly discovered nematode parasite of the highly invasive yellow-legged Asian hornet in France. Using morphological and molecular data the authors place this nematode in the genus Pheromermis and suggest that given its apparent rarity and its paratenic lifecycle, this nematode is very unlikely to be a successful bio-control agent of the yellow-legged Asian hornet in France.

This is a very interesting study and the manuscript is well written. I have only one concern, which is the assertion that these nematodes are exceptionally rare. Based on the presented data, I don’t think this can be confidently stated. As I understand it (though I may be mistaken), in order to detect an internal parasite like a nematode, the host would need to be dissected. However, as the methods are currently written it is unclear whether the 33,000 hornets examined were dissected (Lines 67-69). Likewise the large number of nests dropped and destroyed without noticing any parasites is unconvincing, as these nests are not examined for parasites (Lines 151-153).
If the 33,000 wasps studied were indeed dissected, or there is some other compelling reason to think that a nematode infection would be noticeable without dissection it needs to be clearly stated. If not then the authors should to be more cautious in their discussion of this parasite’s likelihood.

Line 45, change “and resulting in an increase of growth” to “and resulting in increased growth”

Lines 62-63, change “database maintained by the MNHN and regularly updated by one of us, Q.R.” to “database maintained and regularly updated by MNHN”

Lines 67-69, how many districts were the 77 nests from and were there any from the districts the nematode has been found in?

Line 81, replace phrase “one of us” with institute name

Lines 82-85, move to Results section

Lines 80-82, describe what morphological measurements you did

Lines 164-167, does the nematode kill its host when it leaves? If so, say so here

Line 173, change “possible” to “probable”

Lines 184-186, were caddisflies the only potential paratenic hosts listed in lines 180-182 that were found to be part of the wasp prey spectrum in this study? If so then state so explicitly.

Line 190, what does this “50% ratio” refer to?

Line 202, change “does not enable the nematode to greatly infest a colony” to “would not enable the nematode to infest a colony at high densities”

Line 206, Why can’t the founding queen be infected? I don’t understand what over-wintering (or lack thereof) has to do with infection potential

Lines 229-230, Pheromermis should be included in this list

Lines 238-242, are you suggesting this as a control option?

Reviewer 2 ·

Basic reporting

No comments here

Experimental design

No comments here

Validity of the findings

No comments here

Additional comments

I found this article to be well written and containing important and timely information on the subject of the invasive hornet, Vespa velutina. I support the authors’ overall interpretation of their findings and believe this to be acceptable for publication. Please find below some of my minor edits and suggestions.

Introduction
Line 40: Suggest changing “future movements are even more pessimistic” to something along the lines of “future range expansion may be even more rapid than in the past 10 years”.

Materials and methods

Line 108: please change “that” to “than”.

Line 109: Please update the accession number.

Discussion

Line 172: Suggest rewording “single one” to “single individual”.

---

## Round 0.2 · accepted · Accept

Dear Claire, I am writing to inform you that your manuscript has been accepted to PeerJ.